# Prevalence and determinants of mother and newborn skin-to-skin contact in Ghana

**Richard Gyan Aboagye**[1,2]\*, **Khadijat Adeleye**[3], **Bright Opoku Ahinkorah**[4,5]

**1** Department of Family and Community Health, Fred N. Binka School of Public Health, University of Health and Allied Sciences, Hohoe, Ghana, **2** School of Population Health, University of New South Wales, Sydney, New South Wales, Australia, **3** Elaine Marieb College of Nursing, University of Massachusetts Amherst, United States of America, **4** REMS Consultancy Services Limited, Sekondi-Takoradi, Western Region, Ghana, **5** Faculty of Health and Medical Sciences, The University of Adelaide, Adelaide, Australia

\* aboagyegyan94@gmail.com

## Abstract

### Background

Despite the well-established role of skin-to-skin contact in reducing neonatal mortality, its implementation varies significantly across geographical regions, particularly in sub-Saharan Africa. Therefore, we estimated the prevalence of mother and newborn skin-to-skin contact at birth and investigated the factors associated with its practice in Ghana.

### Methods

We used data from the 2022 Ghana Demographic and Health Survey. The analysis included 3833 mother-child pairs. Data was analysed using Stata 17.0, with percentages and confidence intervals (CI) used to present the prevalence of mother and newborn skin-to-skin contact. We employed multilevel binary logistic regression models to examine factors associated with the practice of mother and newborn skin-to-skin contact.

### Results

Overall, 67.2% [64.9 - 69.4] of mothers practised skin-to-skin contact. Delivery by caesarean section was associated with a decreased likelihood of mother and newborn skin-to-skin contact (adjusted odds ratio [aOR] = 0.04; 95% CI: 0.02, 0.06). Higher birth order (fifth or more) (aOR = 2.34, 95%CI: 1.13, 4.84) was significantly associated with increased odds of skin-to-skin contact. Women who had eight or more antenatal care visits were more likely to engage in skin-to-skin contact (aOR = 1.82; 95% CI: 1.04–3.21) than those with fewer than four visits. Women who delivered in a healthcare facility were more likely to practise skin-to-skin contact (aOR = 30.67; 95% CI: 18.93, 49.70) than those who delivered at home. Compared

**Data availability statement:** The dataset used can be accessed via the MEASURE DHS repository https://dhsprogram.com/data/dataset/Ghana_Standard-DHS_2022.cfm?flag=1.

**Funding:** The author(s) received no specific funding for this work.

**Competing interests:** The authors declare that they have no competing interests.

**Abbreviations:** aOR, Adjusted Odds Ratios; AIC, Akaike Information Criterion; CHPS, Community-Based Health Planning and Services; CI, Confidence Interval; DHS, Demographic and Health Survey; ICC, Intra-Class Correlation Coefficient; MEASURE DHS, Monitoring and Evaluation to Assess and Use Results Demographic and Health Surveys; NGOs, Non-governmental Organizations; PSU, Primary Sampling Unit; SDG, Sustainable Development Goal; WHO, World Health Organization.

to women in the Western region, those in the Eastern (aOR = 2.85, 95%CI: 1.21, 6.73), Western North (aOR = 3.87, 95%CI: 1.60, 9.37), Ahafo (aOR = 3.09, 95%CI: 1.19, 8.02), North East (aOR = 4.44, 95%CI: 1.88, 10.50), Upper East (aOR = 3.67, 95%CI: 1.45, 9.31), and Upper West regions (aOR = 6.13, 95%CI: 2.33, 16.16) were more likely to practise skin-to-skin contact.

## Conclusion

Our study has shown moderate mother and newborn skin-to-skin contact practise in Ghana, with significant regional variations. Going forward, any initiatives by the Ghana Health Service, the Ministry of Health, or other organisations focused on maternal and neonatal health must consider the geographical context of their efforts and programme implementation. Enhancing skin-to-skin contact requires increased advocacy and health education during antenatal care sessions, alongside higher attendance at such visits. Additionally, advocating for hospital births and reducing the number of home births is likely to boost skin-to-skin contact practices in Ghana.

## Introduction

Reducing neonatal mortality is a crucial component of the United Nations' Sustainable Development Goal (SDG) 3, which aims to ensure healthy lives and promote well-being for all ages. Specifically, SDG target 3.2 aims to end preventable deaths of newborns and children under five years of age by 2030, with all countries striving to reduce neonatal mortality to at least 12 per 1,000 live births [1]. Despite significant progress in reducing child mortality rates, the neonatal mortality rate in Ghana remains relatively high at 25 deaths per 1,000 live births as of 2020 [2,3].

Neonatal mortality in Ghana remains a significant public health issue [3]. The World Health Organization (WHO) has identified essential newborn care practices as critical interventions to reduce neonatal mortality rates [4]. Among these core interventions are thermoregulation, which includes skin-to-skin contact, drying and wrapping the baby, and delayed bathing; early initiation of breastfeeding; and appropriate cord care [5]. Despite the proven effectiveness of skin-to-skin contact in improving neonatal outcomes, the rate of implementation in Ghana remains understudied.

Mother and newborn skin-to-skin contact, which forms part of the broader kangaroo care, is the practice of placing a naked newborn baby on the mother's bare chest immediately after birth [6]. This practice promotes breastfeeding initiation and duration, facilitates mother-infant bonding, and regulates the newborn's body temperature, heart rate, and respiration [7,8]. It also helps colonise the newborn's skin with maternal commensal bacteria, which may play a role in developing the infant's immune system [9]. Additionally, it has been associated with reduced stress and improved pain management, empowering mothers to take an active role in their newborn's care [10,11].

Despite the well-established benefits, the practice of skin-to-skin contact varies widely across regions, particularly in sub-Saharan Africa. Aboagye et al. [12] reported an overall prevalence of 42% for skin-to-skin contact in sub-Saharan Africa. However,

the rates varied substantially across countries, ranging from 11.7% in Nigeria to 75.1% in Benin. Similarly, Dirirsa et al. [13] found that only 44.8% of mothers practised skin-to-skin contact within the first hour after birth in Ethiopia.

Several factors can influence the practice of skin-to-skin contact. These factors include healthcare provider knowledge and attitudes, hospital policies and guidelines [13,14], cultural beliefs and medical concerns [15], and logistical barriers [16,17]. In the Ghanaian context, Saaka et al. [18] highlighted the need for healthcare provider training and education to address knowledge gaps in essential newborn practices.

As the benefits of skin-to-skin contact become increasingly recognised globally, it is essential to continue monitoring its prevalence in Ghana and identify strategies to overcome barriers to its widespread adoption, ultimately improving maternal and infant health outcomes in the country and contributing to the achievement of SDG target 3.2. To the best of our knowledge, after an extensive literature search, no study has been conducted in Ghana to investigate skin-to-skin contact using nationally representative data. Previous studies were conducted in sub-Saharan Africa [7,12], Nigeria [15], Ethiopia [13], and Uganda [16]. This leaves a gap in the literature, which our study seeks to fill. Therefore, this study aimed to (i) estimate the prevalence of mother and newborn skin-to-skin contact in Ghana and (ii) examine the factors associated with the practice.

## Methods

### Overview of the 2022 Ghana Demographic and Health Survey

We used data from the 2022 Ghana Demographic and Health Survey (DHS), the seventh DHS since its inception in 1988 [19,20]. DHS is a nationwide survey conducted in over 90 low- and middle-income countries to understand health and demographic issues, including maternal and newborn skin-to-skin contact [20]. In Ghana, the 2022 DHS was conducted across all 16 regions, with data collected from both rural and urban areas [19]. A detailed description of the DHS methodology has been published in the literature [21]. Briefly, a cross-sectional design was adopted for the Ghana DHS, and respondents, including men, women, and children, were sampled using a multistage sampling method [19]. A stratified two-stage cluster sampling was used in the 2022 Ghana DHS, which produced a representative sample at the national level, for urban and rural areas, and for each of the country's 16 regions. In the first stage, 618 clusters were selected from the sampling frame using a probability-proportional-to-size method for both urban and rural areas in each region. Subsequently, the clusters were chosen with equal probability through systematic random sampling from those selected in the first phase for both urban and rural areas. In the second stage, after the clusters had been selected, a household listing and map-updating process was carried out in all selected clusters to create a household list for each. This list was then used as a sampling frame to select the household sample, with 30 households in each cluster randomly chosen from the list for interviews. In the surveyed households, 15,317 women aged 15–49 were identified as eligible for individual interviews. Interviews were conducted with 15,014 women, yielding a response rate of 98.0% [19]. Of these, 6,839 had a history of pregnancy before the survey. In this study, 3833 mother-child pairs were included in the analysis. This sample represents the number of mother-child pairs with complete observations on variables of interest included in the study. This paper was written by adhering to the Strengthening the Reporting of Observational Studies in Epidemiology (STROBE) checklist for reporting guidelines [22].

### Variables

The outcome variable was the mother and newborn skin-to-skin contact. The question "*Was child put on mother's chest and bare skin after birth?*" was used to assess the practice of skin-to-skin contact in the DHS. Response categories to this question were 'no', 'put on the chest, touching bare skin'; 'put on the chest, no touching of bare skin'; put on the chest, don't know/missing on touching of bare skin', and 'don't know'. We coded the option "put on chest, touching the bare skin" as '1=yes', while the remaining responses were coded as 0 = no [7,12,23].

Nineteen (19) exposure variables were included in the study. These variables were selected from the literature [7,12,23] and are available in the DHS dataset. With reference to previous studies that used the DHS dataset [7,12,23], we grouped the variables into individual and contextual levels. The individual level variables consisted of sex of the child, birth order, size of child at birth (qualitative self-report from the mother), caesarean section delivery, type of birth, mother's age, level of education, marital status, current working status, wanted last pregnancy, number of antenatal care visits, place of delivery, and mass media exposure (watching television, listening to the radio, reading newspapers or magazines, and internet usage). The contextual-level variables were household wealth index, place of residence, and region. Table 1 shows the categories of each explanatory variable included in the study.

## Statistical analyses

Data was cleaned and analysed using Stata version 17.0. The prevalence and regional variations in skin-to-skin contact were presented using percentages and a spatial map (Fig 1). Next, a bivariate analysis was performed to examine the distribution of skin-to-skin contact across the exposure variables. We presented the results of the bivariate analysis as percentages with 95% confidence intervals (CIs). Subsequently, the best variable selection method was adopted to select the best-fitted variable for the regression analysis [24,25]. The aggregate of variables with the lowest Akaike Information Criterion value was selected as the best-fitting variables for the regression analysis. As a result, a total of nine (9) variables: sex of child, size of child at birth, type of birth, household wealth index, place of residence, and mass media exposure (watching television, listening to the radio, reading newspapers or magazines, and internet usage) were dropped. Before the regression analysis, we checked for evidence of multicollinearity among the selected variables. The results showed that the minimum, maximum, and mean variance inflation factor values were 1.01, 2.18, and 1.32, respectively. Hence, there was no evidence of high collinearity among the variables. Later, we performed a four-modelled multilevel binary logistic regression analysis to examine the factors associated with mother and newborn skin-to-skin contact. The first model (Model I) had no explanatory variables and was considered the empty model, which accounted for the variation in mother and newborn skin-to-skin contact attributable to clustering at the primary sampling unit. We included the individual- and contextual-level variables in Models II and III, respectively. Model IV contained all the explanatory variables.

Two results were derived from the regression analysis: the fixed-effect and random-effect results. The fixed effect results examine the association between the explanatory variables and mother and newborn skin-to-skin contact. The results were presented as adjusted odds ratios (aOR) with their respective 95% CI. The random effect explained variation in mother and newborn skin-to-skin contact across the models. The intra-cluster correlation coefficient (ICC) was used to assess the extent of variation in the mother and newborn skin-to-skin contact across the four models. The ICC values indicated the extent to which variation in the mother and newborn skin-to-skin contact was due to differences between clusters. Zero ICC values showed no evidence of variation between clusters, whereas high ICC values indicated greater variation between clusters. All analyses were weighted, and the Stata survey dataset function "*svy*" was used in all analyses. All missing data were dropped during data cleaning before generating the results. Statistical significance was set at p < 0.05 in the regression analysis.

## Ethical consideration

Given the reliance on a secondary dataset (the Ghana DHS) for this study, no ethical clearance was sought. Before using the dataset, permission to use the Ghana DHS was obtained from the Monitoring and Evaluation to Assess and Use Results DHS (MEASURE DHS).

## Results

### Prevalence of mother and newborn skin-to-skin contact practise in Ghana

Fig 1 shows the regional distribution of the prevalence of mothers who practised skin-to-skin contact with their newborns. The regions with the highest prevalence were Upper West (85.5%), North East (82.4%), Upper East (82.1%), and Ahafo

**Table 1. Bivariate analysis of mother and newborn skin-to-skin contact.**

| Variables | Sample (percentage) n (%) | Practised Skin-to-skin contact % [95% CI] |
|---|---|---|
| **Prevalence** | | **67.2 [64.9,69.4]** |
| **Sex of child** | | |
| Male | 1,982 (51.7) | 67.7 [64.6, 70.6] |
| Female | 1,851(48.3) | 66.7 [63.6, 69.6] |
| **Birth order** | | |
| First | 896 (23.4) | 67.1 [62.6, 71.3] |
| Second | 787 (20.5) | 67.5 [63.0, 71.7] |
| Third | 602 (15.7) | 69.4 [64.3, 74.0] |
| Fourth | 527 (13.8) | 68.1 [62.4, 73.3] |
| Fifth or more | 1,021 (26.6) | 65.2 [60.6, 69.6] |
| **Size of child at birth** | | |
| Very large | 529 (13.8) | 65.8 [59.6, 71.6] |
| Larger than average | 1,180 (30.8) | 67.1 [63.1, 70.8] |
| Average | 1,647 (43.0) | 69.2 [66.0, 72.3] |
| Smaller than average | 327 (8.5) | 64.1 [57.3, 70.5] |
| Small | 149 (3.9) | 56.7 [46.3, 66.6] |
| **Delivery by caesarean section** | | |
| No | 3,068 (80.0) | 74.9 [72.2, 77.4] |
| Yes | 765 (20.0) | 36.2 [32.0, 40.7] |
| **Type of birth** | | |
| Single | 3,731 (97.3) | 67.6 [65.2, 69.9] |
| Multiple | 102 (2.7) | 51.1 [38.4, 63.5] |
| **Mother's age (years)** | | |
| 15-24 | 1,103 (28.8) | 67.1 [63.0, 71.0] |
| 25-34 | 1,841 (48.0) | 69.7 [66.6, 72.6] |
| 35 and above | 889 (23.2) | 62.0 [57.6, 66.3] |
| **Level of education** | | |
| No education | 798 (20.8) | 65.6 [59.7, 71.0] |
| Primary | 594 (15.5) | 61.9 [56.3, 67.1] |
| Secondary | 2,085 (54.4) | 69.5 [66.6, 72.3] |
| Higher | 356 (9.3) | 65.7 [58.7, 72.1] |
| **Marital status** | | |
| Never married | 498 (13.0) | 60.1 [53.9, 66.0] |
| Married | 2,300 (60.0) | 71.1 [68.1, 73.9] |
| Cohabiting | 861 (22.5) | 61.8 [57.3, 66.1] |
| Previously married | 174 (4.5) | 62.3 [51.8, 71.8] |
| **Current working status** | | |
| Not working | 924 (24.1) | 68.8 [64.9, 72.5] |
| Working | 2,909 (75.9) | 66.6 [63.8, 69.4] |
| **Wanted the last pregnancy** | | |
| Wanted then | 2,295 (59.8) | 70.3 [67.4, 73.0] |
| Wanted later | 1,179 (30.8) | 63.2 [59.3, 66.9] |
| Wanted no more | 359 (9.4) | 60.4 [52.7, 67.6] |

*(Continued)*

**Table 1.** (Continued)

| Variables | Sample (percentage) n (%) | Practised Skin-to-skin contact % [95% CI] |
|---|---|---|
| **Antenatal care visits** | | |
| Below four visits | 467 (12.2) | 50.0 [43.1, 56.8] |
| Four to seven visits | 1,884 (49.1) | 67.6 [64.5, 70.6] |
| Eight or more visits | 1,482 (38.7) | 72.1 [68.7, 75.2] |
| **Place of delivery** | | |
| Home or other | 533 (13.9) | 25.9 [21.4, 31.0] |
| Health facility | 3,300 (86.1) | 73.8 [71.6, 75.9] |
| **Watch television** | | |
| No | 1,143 (29.8) | 65.6 [61.0, 69.9] |
| Yes | 2,690 (70.2) | 67.8 [65.4, 70.2] |
| **Listen to radio** | | |
| No | 1,400 (36.5) | 66.7 [62.8, 70.4] |
| Yes | 2,433 (63.5) | 67.4 [64.7, 70.0] |
| **Read newspapers or magazines** | | |
| No | 3,517 (91.8) | 66.7 [64.3, 69.1] |
| Yes | 316 (8.2) | 72.0 [65.2, 78.0] |
| **Internet usage** | | |
| No | 2,363 (61.7) | 67.1 [64.1, 70.0] |
| Yes | 1,470 (38.3) | 67.2 [63.6, 70.7] |
| **Wealth index** | | |
| Poorest | 934 (24.4) | 62.2 [56.8, 67.2] |
| Poorer | 793 (20.7) | 70.5 [65.8, 74.8] |
| Middle | 775 (20.2) | 70.6 [65.6, 75.2] |
| Richer | 693 (18.1) | 65.8 [60.6, 70.7] |
| Richest | 637 (16.6) | 67.6 [61.7, 73.0] |
| **Place of residence** | | |
| Urban | 1,784 (46.5) | 68.1 [64.9, 71.1] |
| Rural | 2,049 (53.5) | 66.4 [63.0, 69.5] |
| **Region** | | |
| Western | 226 (5.9) | 65.3 [59.0, 71.2] |
| Central | 394 (10.3) | 60.5 [53.2, 67.4] |
| Greater Accra | 451 (11.8) | 63.1 [55.4, 70.2] |
| Volta | 143 (3.7) | 65.4 [57.5, 72.6] |
| Eastern | 271 (7.1) | 72.0 [64.3, 78.5] |
| Ashanti | 695 (18.1) | 61.2 [54.7, 67.3] |
| Western North | 106 (2.8) | 75.4 [66.6, 82.5] |
| Ahafo | 83 (2.2) | 77.5 [70.2, 83.5] |
| Bono | 124 (3.2) | 73.9 [64.7, 81.4] |
| Bono East | 210 (5.5) | 66.7 [60.0, 72.8] |
| Oti | 135 (3.5) | 56.6 [48.4, 64.5] |
| Northern | 433 (11.3) | 65.6 [55.0, 74.8] |
| Savannah | 115 (3.0) | 71.9 [62.5, 79.7] |
| North East | 124 (3.2) | 82.4 [76.1, 87.4] |
| Upper East | 208 (5.4) | 82.1 [75.6, 87.2] |
| Upper West | 115 (3.0) | 85.5 [79.4, 90.0] |

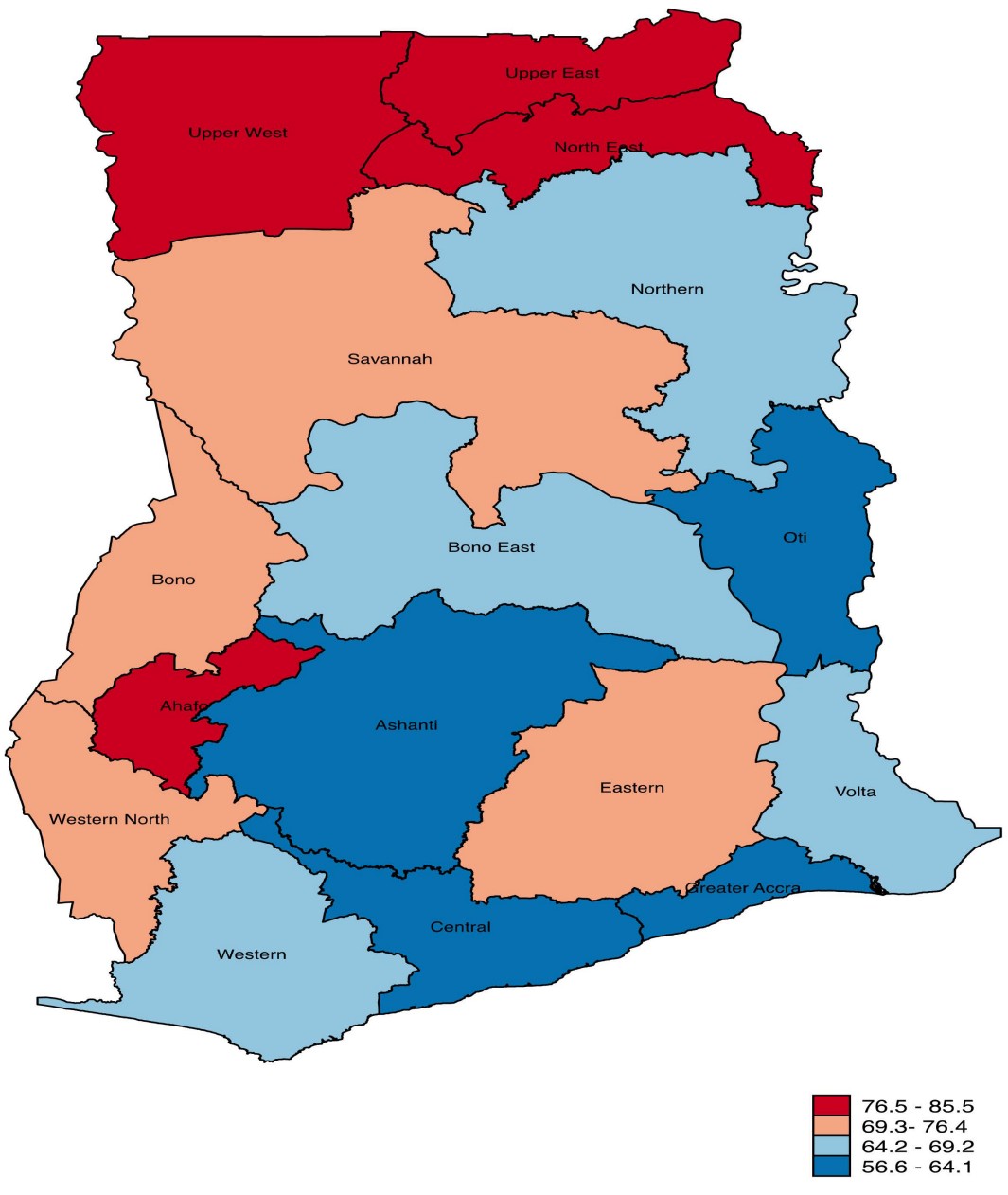

**Fig 1. Prevalence of mother and newborn skin-to-skin contact practise in Ghana.**

(77.5%), as indicated by the red-shaded areas on the map. Greater Accra (63.3%), Ashanti (61.2%), Central (60.5%), and Oti (56.6%) had the lowest prevalence of skin-to-skin contact. Overall, 67.2% [64.9 - 69.4] of mothers practised skin-to-skin contact (Table 1).

**Bivariate analysis of mother and newborn skin-to-skin contact practice across the explanatory variables**

Table 1 presents the distribution of skin-to-skin contact across the explanatory variables. Overall, the prevalence of skin-to-skin contact was 67.2% [64.9,69.4]. Across birth order, the highest prevalence of mother and newborn skin-to-skin

contact was observed among third-order newborns (69.4%), whereas the lowest was among those in the fifth or higher order (65.2%). The highest prevalence was observed in male children (67.7%) and in those of average birth size (69.2%). Higher practise of skin-to-skin contact was observed among women who did not deliver by caesarean section (74.9%), mothers aged 25–34 (69.7%), and those with secondary education (69.5%). Mother and newborn skin-to-skin contact was practised more by married women (71.1%), those with eight or more antenatal care visits (72.1%), and those residing in the Upper West (85.5%) (Table 1).

### Factors associated with mother and newborn skin-to-skin contact

**Fixed effect results.** Table 2 presents the factors associated with skin-to-skin contact in Ghana. The results show that delivery by caesarean section was significantly associated with a decreased likelihood of skin-to-skin contact (aOR = 0.04, 95%CI: 0.02, 0.06). Higher birth order (fifth or more) (aOR = 2.34, 95%CI: 1.13, 4.84) was significantly associated with increased odds of skin-to-skin contact. Furthermore, women who had eight or more antenatal care visits were significantly more likely to engage in skin-to-skin contact (aOR = 1.82, 95% CI: 1.04, 3.21) than those with fewer than four visits. Delivery in a healthcare facility was associated with a higher likelihood of practising skin-to-skin contact (aOR = 30.67; 95%CI: 18.93, 49.70) compared to those who deliver at home. Compared to women in the Western region, those in the Eastern (aOR = 2.85, 95%CI: 1.21, 6.73), Western North (aOR = 3.87, 95%CI: 1.60, 9.37), Ahafo (aOR = 3.09, 95%CI: 1.19, 8.02), North East (aOR = 4.44, 95%CI: 1.88, 10.50), Upper East (aOR = 3.67, 95%CI: 1.45, 9.31), Upper West (aOR = 6.13, 95%CI: 2.33, 16.16), Bono (aOR = 3.12; 95% CI: 1.28, 7.59), Northern (aOR = 2.31; 95% CI: 1.02, 5.25), and Savannah (aOR = 2.88; 95% CI: 1.26, 6.59) were more likely to practise skin-to-skin contact.

### Random effect results

As shown in Table 2, there were variations in the mother and newborn skin-to-skin contact due to differences between the clusters. The variation in mother and newborn skin-to-skin contact was 0.48 [0.41,0.54] in Model I, indicating that 48% of the variation in mother and newborn skin-to-skin contact was due to the difference between the clusters, and the remaining 52.0% was due to the differences within the clusters. The ICC value increased to 0.51 [0.44, 0.58] in Model II. It was then reduced to 0.45 [0.38, 0.52] in Model III and increased again to 0.48 [0.41, 0.55] in the final model. Differences in primary sampling unit (PSU) values were also observed across the four models, with the highest PSU values in Model II (3.39 [2.58, 4.46]). In Model II, the PSU value, along with its corresponding ICC, indicated that where a woman lived (PSU) accounted for 51.0% of the differences in mother and newborn skin-to-skin contact before adjusting for individual-level variables.

## Discussion

Existing evidence from Ghana has established that neonates who are deprived of skin-to-skin contact immediately after birth are more likely to die within the neonatal period compared to those who benefit from skin-to-skin contact [26]. Understanding the nuances of mother and newborn skin-to-skin contact is a public health imperative. For this reason, we estimated the prevalence and examined the factors associated with mother and newborn skin-to-skin contact in Ghana using the 2022 DHS data. Our findings revealed that more than half of the mothers in Ghana (67.2%) practised skin-to-skin contact. The estimated prevalence is higher than that reported in Papua New Guinea (45.2%) [23], the Gambia (35.7%) [27], and sub-Saharan Africa (42%) [12]. Also, the prevalence of mother and newborn skin-to-skin contact in our study is higher than that reported in Nigeria, where 12% of mothers practised skin-to-skin contact [15]. The relatively high prevalence of mother and newborn skin-to-skin contact in Ghana is an encouraging indicator of the effectiveness of healthcare initiatives and policies aimed at promoting maternal and newborn wellbeing. This practice reflects the successful efforts of governmental bodies such as the Ministry of Health and the Ghana Health Service, as well as supportive policies, including health insurance coverage and the free maternal healthcare policy [28,29].

**Table 2. Factors associated with mother and newborn skin-to-skin contact.**

| Variable | Model I | Model II aOR [95% CI] | Model III aOR [95% CI] | Model IV aOR [95% CI] |
|---|---|---|---|---|
| **Fixed effects results** | | | | |
| **Delivery by caesarean section** | | | | |
| No | | 1.00 | | 1.00 |
| Yes | | 0.04*** [0.02, 0.06] | | 0.04*** [0.02, 0.06] |
| **Birth order** | | | | |
| First | | 1.00 | | 1.00 |
| Second | | 1.25 [0.76, 2.07] | | 1.26 [0.76, 2.07] |
| Third | | 1.52 [0.90, 2.58] | | 1.54 [0.91, 2.59] |
| Fourth | | 1.84 [0.97, 3.49] | | 1.87 [0.99, 3.54] |
| Fifth or more | | 2.31*[1.11, 4.77] | | 2.34* [1.13, 4.84] |
| **Mother's age (years)** | | | | |
| 15-24 | | 1.00 | | 1.00 |
| 25-34 | | 0.97 [0.61, 1.56] | | 0.98 [0.61, 1.57] |
| 35 and above | | 0.54 [0.27, 1.06] | | 0.54 [0.27, 1.07] |
| **Marital status** | | | | |
| Never married | | 1.00 | | 1.00 |
| Married | | 1.60 [0.86, 3.00] | | 1.54 [0.83, 2.89] |
| Cohabiting | | 1.02 [0.56, 1.88] | | 1.02 [0.56, 1.88] |
| Previously married | | 1.86 [0.78, 4.42] | | 1.85 [0.78, 4.40] |
| **Level of education** | | | | |
| No education | | 1.00 | | 1.00 |
| Primary | | 1.00 [0.61, 1.63] | | 1.02 [0.62, 1.69] |
| Secondary | | 1.27 [0.82, 1.96] | | 1.32 [0.85, 2.06] |
| Higher | | 1.26 [0.60, 2.67] | | 1.34 [0.63, 2.85] |
| **Current working status** | | | | |
| Not working | | 1.00 | | 1.00 |
| Working | | 0.74 [0.51, 1.07] | | 0.75 [0.52, 1.09] |
| **Wanted the last pregnancy** | | | | |
| Wanted then | | 1.00 | | 1.00 |
| Wanted later | | 0.81 [0.53, 1.24] | | 0.82 [0.54, 1.26] |
| Wanted no more | | 0.78 [0.39, 1.53] | | 0.79 [0.40, 1.55] |
| **Antenatal care visits** | | | | |
| Below four visits | | 1.00 | | 1.00 |
| Four to seven visits | | 1.63 [0.97, 2.76] | | 1.63 [0.96, 2.76] |
| Eight or more visits | | 1.81* [1.03, 3.18] | | 1.82* [1.04, 3.21] |
| **Place of delivery** | | | | |
| Home or other | | 1.00 | | 1.00 |
| Health facility | | 30.58***[18.91, 49.46] | | 30.67*** [18.93, 49.70] |
| **Region** | | | | |
| Western | | | 1.00 | 1.00 |
| Central | | | 1.09[0.54, 2.18] | 1.21 [0.53, 2.78] |
| Greater Accra | | | 1.59[0.74, 3.42] | 1.85 [0.84, 4.07] |
| Volta | | | 1.37[0.60, 3.09] | 1.86 [0.74, 4.71] |
| Eastern | | | 2.14*[1.01, 4.54] | 2.85* [1.21, 6.73] |

*(Continued)*

**Table 2.** (Continued)

| Variable | Model I | Model II<br>aOR [95% CI] | Model III<br>aOR [95% CI] | Model IV<br>aOR [95% CI] |
|---|---|---|---|---|
| Ashanti | | | 0.94[0.47, 1.87] | 0.83 [0.38, 1.82] |
| Western North | | | 3.22**[1.39, 7.45] | 3.87** [1.60, 9.37] |
| Ahafo | | | 3.08**[1.43, 6.66] | 3.09* [1.19, 8.02] |
| Bono | | | 2.82**[1.32, 6.04] | 3.12* [1.28, 7.59] |
| Bono East | | | 1.61[0.78, 3.32] | 2.10 [0.96, 4.62] |
| Oti | | | 0.82[0.44, 1.56] | 1.27 [0.62, 2.59] |
| Northern | | | 1.78[0.85, 3.71] | 2.31* [1.02, 5.25] |
| Savannah | | | 2.14*[1.04, 4.41] | 2.88* [1.26, 6.59] |
| North East | | | 4.42***[2.14, 9.11] | 4.44*** [1.88, 10.50] |
| Upper East | | | 4.35***[2.02, 9.37] | 3.67** [1.45, 9.31] |
| Upper West | | | 5.73***[2.59, 12.69] | 6.13*** [2.33, 16.16] |
| **Random effect model** | | | | |
| PSU variance (95% CI) | 3.01 [2.30, 3.93] | 3.39 [2.58, 4.46] | 2.66 [2.07, 3.12] | 3.05 [2.31, 4.02] |
| ICC | 0.48 [0.41,0.54] | 0.51 [0.44, 0.58] | 0.45 [0.38, 0.52] | 0.48 [0.41, 0.55] |
| N | 3833 | 3833 | 3833 | 3833 |
| Number of clusters | 616 | 616 | 616 | 616 |

*The results in the fixed effect are adjusted odds ratios, and the numbers in the parentheses represent the 95% confidence intervals

* p < 0.05, ** p < 0.01, *** p < 0.001; 1.00 = Reference category; PSU = Primary Sampling Unit; ICC = Intra-Class Correlation Coefficient

The prevalence of skin-to-skin contact varied significantly by region of residence, with mothers in the Upper West region reporting the highest proportions and likelihood of practising it. The spatial analysis also revealed that the practice of mother and newborn skin-to-skin contact is higher in the Savannah zone (Upper West, Northern, and Upper East) than in the coastal and middle zones. The Savannah regions, for example, the Upper West region, are primarily rural in nature [30]; women from this region tend to be very traditional and more inclined to follow traditional practices. Culturally, skin-to-skin contact is considered part of the region's childbirth practices. Hence, this may account for the high mother and newborn skin-to-skin contact practise in the region. The higher practice of skin-to-skin contact in the Savannah zone compared to the coastal and middle zones could be due to the implementation of the Community-Based Health Planning and Services (CHPS) projects, the concentration of maternal and child health-related non-governmental organisations (NGOs) and advocacy interventions in the Savannah zone. These NGOs often prioritise regions with higher maternal and child health challenges, deploying targeted interventions to improve practices such as skin-to-skin contact. However, we recommend conducting qualitative and regional studies to explore the cultural factors influencing regional differences in the practice of mother and newborn skin-to-skin contact.

Consistent with previous literature [12,27], our study showed that birth by caesarean section was significantly associated with a lower likelihood of practising skin-to-skin contact. Medical procedures and protocols surrounding cesarean births (e.g., anaesthesia and medical monitoring) may limit or delay the immediate initiation of skin-to-skin contact compared to vaginal births. It is also possible that the recovery process for mothers who undergo cesarean sections might involve more extended hospital stays or postoperative discomfort, anxiety and distress [31], all of which can impede their ability to engage in skin-to-skin contact.

Mothers who delivered in a healthcare facility and those who had more than eight antenatal care visits were more likely to practice skin-to-skin contact. Similar findings have been reported in Papua New Guinea [23], Gambia [27] and

sub-Saharan Africa [12]. High antenatal care visits expose mothers to health information about their pregnancy, including the need to practice skin-to-skin contact [27]. This heightened health literacy, associated with higher utilisation of maternal healthcare services, is likely to enhance women's self-efficacy in practising skin-to-skin contact.

Consistent with studies from Nigeria [15] and sub-Saharan Africa [12], we found that birth order significantly predicts mothers' practice of skin-to-skin contact with their newborns. Specifically, higher birth order was positively associated with skin-to-skin contact. Our finding implies that mothers may become more comfortable and confident in practising this beneficial bonding technique with subsequent children. Experienced mothers may have a better understanding of the benefits of practising skin-to-skin contact and are more likely to prioritise it based on their past experiences with other children.

### Implications for policy and practice

The regional variation in skin-to-skin contact practices underscores the importance of considering contextual factors and cultural norms in healthcare interventions. Also, the association between birth by cesarean section and reduced likelihood of skin-to-skin contact underscores the importance of addressing barriers to immediate contact in surgical births, such as prolonged hospital stays and postoperative discomfort. The Ghana Health Service and the Ministry of Health must develop a protocol or guideline to expedite the initiation of skin-to-skin contact for mothers who deliver by caesarean section. Our findings highlight the need to leverage antenatal care sessions to promote skin-to-skin contact between mothers and their newborns. Encouraging more antenatal attendance is likely to improve self-efficacy and health literacy, promoting mothers' skin-to-skin contact practise.

### Strengths and limitations

A key strength of our findings lies in the data used. The Ghana DHS is representative of the country at the district, regional and national levels. Additionally, the use of multilevel modelling was appropriate and increased the validity of our findings by illustrating both fixed and random effects. Nevertheless, the cross-sectional nature of the DHS dataset limits our study's ability to establish causal associations. Also, the Ghana DHS used a cross-sectional design; Hence, we cannot generalise our findings to all mothers in Ghana. Further, the cross-sectional design limited the study to exploring variables (factors) such as cultural beliefs and assertions that may have influenced mother and newborn skin-to-skin contact. It would be interesting to know if the practice of skin-to-skin contact varies for minority groups like women living with disabilities; however, the GDHS does not provide data on that. Additionally, there is no specific quantitative or qualitative data available in the Ghana DHS to explain regional variations in the prevalence of mother and newborn skin-to-skin contact. Thus, limiting the extent of our discussion. Moreover, the exceptionally high odds ratios for some variables, such as facility delivery, and the low odds ratios for cesarean section delivery should be interpreted with caution, as they may reflect confounding variables or data distribution effects. Also, the DHS used self-reports to measure mother and newborn skin-to-skin contact, which are prone to recall bias.

### Conclusion

Our study has shown moderate mother and newborn skin-to-skin contact practise in Ghana, with variations across the regions. Moving forward, any programme or interventions by the Ghana Health Service, the Ministry of Health, or other institutions focused on maternal and neonatal health must factor in the geographical context of their interventions and programme implementation. Improving mother and newborn skin-to-skin contact would require greater advocacy and health education during antenatal care sessions, alongside increased attendance at antenatal care sessions. Also, reducing the number of home births is likely to increase the mother and newborn skin-to-skin contact practice in Ghana.

## Author contributions

**Conceptualization:** Richard Gyan Aboagye, Khadijat Adeleye, Bright Opoku Ahinkorah.

**Data curation:** Richard Gyan Aboagye, Bright Opoku Ahinkorah.

**Formal analysis:** Richard Gyan Aboagye, Bright Opoku Ahinkorah.

**Investigation:** Richard Gyan Aboagye, Khadijat Adeleye, Bright Opoku Ahinkorah.

**Methodology:** Richard Gyan Aboagye, Khadijat Adeleye, Bright Opoku Ahinkorah.

**Resources:** Richard Gyan Aboagye, Bright Opoku Ahinkorah.

**Software:** Richard Gyan Aboagye, Bright Opoku Ahinkorah.

**Supervision:** Richard Gyan Aboagye, Bright Opoku Ahinkorah.

**Validation:** Richard Gyan Aboagye, Khadijat Adeleye, Bright Opoku Ahinkorah.

**Visualization:** Richard Gyan Aboagye, Bright Opoku Ahinkorah.

**Writing – original draft:** Richard Gyan Aboagye, Khadijat Adeleye, Bright Opoku Ahinkorah.

**Writing – review & editing:** Richard Gyan Aboagye, Khadijat Adeleye, Bright Opoku Ahinkorah.

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
