## [Decision Letter · Decision Letter 0]

9 Jul 2025

Dear Dr. ABOAGYE,

Thank you for submitting your manuscript to PLOS ONE. After careful consideration, we feel that it has merit but does not fully meet PLOS ONE’s publication criteria as it currently stands. Therefore, we invite you to submit a revised version of the manuscript that addresses the points raised during the review process.

There are some changes to be done and the reviewers have suggested them. They are necessary to improve the manuscript.

We look forward to receiving your revised manuscript.

Kind regards,

Ricardo Q. Gurgel, PhD

Academic Editor

PLOS ONE

Journal Requirements:

2. We note that Figure 1 in your submission contain [map/satellite] images which may be copyrighted. All PLOS content is published under the Creative Commons Attribution License (CC BY 4.0), which means that the manuscript, images, and Supporting Information files will be freely available online, and any third party is permitted to access, download, copy, distribute, and use these materials in any way, even commercially, with proper attribution. For these reasons, we cannot publish previously copyrighted maps or satellite images created using proprietary data, such as Google software (Google Maps, Street View, and Earth). For more information, see our copyright guidelines: http://journals.plos.org/plosone/s/licenses-and-copyright.

3. Please remove all personal information, ensure that the data shared are in accordance with participant consent, and re-upload a fully anonymized data set.

Reviewers' comments:

Reviewer's Responses to Questions

**Comments to the Author**

1. Is the manuscript technically sound, and do the data support the conclusions?

Reviewer #1: Yes

Reviewer #2: Partly

2. Has the statistical analysis been performed appropriately and rigorously?

Reviewer #1: Yes

Reviewer #2: Yes

3. Have the authors made all data underlying the findings in their manuscript fully available?

Reviewer #1: Yes

Reviewer #2: Yes

4. Is the manuscript presented in an intelligible fashion and written in standard English?

Reviewer #1: Yes

Reviewer #2: Yes

Reviewer #1: Thank you for the opportunity to review this manuscript.

Strengths

The title is clear and appropriately descriptive of the study’s focus on mother and newborn skin-to-skin contact (SSC) in Ghana.

The abstract is well-structured with distinct sections (Background, Methods, Results, Conclusion), reporting key prevalence and adjusted odds ratios (aORs) for important predictors such as cesarean delivery, birth order, antenatal care, place of delivery, and regional differences. Keywords are relevant and enhance discoverability.

The introduction effectively situates the study within global health goals (SDG 3), highlights neonatal mortality in Ghana, summarizes SSC benefits comprehensively, and identifies barriers and research gaps clearly.

Use of the 2022 Ghana Demographic and Health Survey (GDHS) ensures nationally representative, high-quality data. The cross-sectional design, multistage sampling, and multilevel logistic regression with survey weights are appropriately described and applied.

Clear presentation of SSC prevalence overall and by region, with detailed descriptive statistics and robust multilevel modeling results including random effects and ICC values.

The discussion contextualizes findings well, compares with other countries, explains associations plausibly, and outlines clear policy implications. Strengths and limitations are acknowledged transparently.

Areas for Improvement

Title

Consider refining to “Prevalence and determinants of mother and newborn skin-to-skin contact in Ghana” to better reflect the study’s dual focus on prevalence and predictors, improving clarity and searchability.

Abstract

The extremely large aOR for facility delivery (30.67) and very low aOR for cesarean section (0.04) should be briefly contextualized to caution readers about potential confounding or data distribution effects.

Rephrase “higher birth order showed an incremental increase…” to “higher birth order (fifth or more) was significantly associated with increased odds…” for accuracy.

Ensure consistency in reported prevalence figures (67.2% vs. 62.7%).

Minor language edits for tense consistency and clarity are recommended.

Consider adding keywords like “Neonatal health,” “Maternal health,” or “Prevalence” to broaden indexing.

Introduction

Improve paragraph structure to separate global context, SSC benefits, prevalence patterns, and barriers for easier reading.

Clarify terminology distinguishing SSC from kangaroo care, as the latter includes SSC plus other components.

Strengthen the statement about no prior nationally representative studies by referencing smaller or regional studies and their limitations.

Make the link between SSC and neonatal mortality more explicit to emphasize SSC’s potential impact in Ghana.

Conclude with a clear statement of study objectives.

Methods

Provide more detail on sampling frame specifics (number of clusters, households per cluster) and response rates for transparency.

Clarify handling of missing data and whether imputation was performed.

Elaborate on variable selection process using AIC, specifying which variables were retained or excluded.

Reconsider classifying household wealth index as a contextual variable, as it is typically household-level.

Include checks for multicollinearity, interactions, and model diagnostics.

Explain the intracluster correlation coefficient (ICC) and its interpretation for readers unfamiliar with multilevel modeling.

Results

Correct inconsistency in overall SSC prevalence figures.

Interpret the large aOR for facility delivery cautiously, discussing possible sparse data bias or confounding.

Avoid claiming “incremental increase” in SSC odds with birth order if only the highest category is significant.

Interpret ICC and PSU variation in terms of practical significance.

Discussion

Avoid causal language given the cross-sectional design; rephrase statements suggesting causality.

Critically examine the unusually high odds ratio for facility delivery.

Discuss potential confounders (e.g., socioeconomic status, education) and biases (e.g., recall bias) more thoroughly.

Correct minor typographical errors (e.g., “SCC” instead of “SSC”).

Recommendations

Standardize prevalence figures throughout the manuscript.

Increase transparency on variable selection, missing data handling, and model diagnostics.

Interpret extreme odds ratios with caution; consider presenting absolute risks or predicted probabilities.

Avoid causal language and clearly state cross-sectional design limitations.

Expand discussion on confounders and biases to strengthen validity.

Suggest qualitative follow-up studies to explore cultural and regional SSC differences.

Proofread carefully to eliminate minor error

Reviewer #2: This manuscript uses representative data from Ghana to study factors associated with mother and newborn skin-to-skin contact. The use of multi-level analysis to consider regional context is a strong point. Please see my detailed feedback below.

1. Abbreviation Clarification in Methods

On page 5, the abbreviation "GDHS" is first introduced. For clarity and academic rigor, I recommend that the full term—Ghana Demographic and Health Survey (GDHS)—be spelled out at its first mention, with the abbreviation in parentheses.

2. Sample Size Before and After Missing Data Exclusion

In the Methods section (page 5), it appears that a complete-case analysis was performed, resulting in a final analytic sample of 3,833 mother-child pairs. Please specify the initial sample size before excluding cases with missing data.

3. Details of Variable Selection and AIC Reporting in Statistical Analysis

On page 6, the Statistical Analyses subsection states that "the best-variable selection method was adopted to select the best-fitted variable for the regression analysis." Although the manuscript mentions the use of the Akaike Information Criterion (AIC) as a criterion for model selection, it does not clearly specify which variables were initially considered in the selection process, nor does it provide the actual AIC values. Please clarify which variables were included in the initial stage, describe the selection method used, and report the AIC values or thresholds that guided the final model choice.

4. Interpretation of Prevalence by Birth Order and Sex

On page 8, the statement that "the highest prevalence was observed among third-born children (69.4%), male children (67.7%), and those in higher birth order (65.2%)" is confusing. Since "third" and "higher birth order" (e.g., fifth or more) are distinct and mutually exclusive categories, only one should represent the true highest value.

5. Calculation of 95% Confidence Interval in Table 1

Table 1 presents the bivariate analysis of mother and newborn skin-to-skin contact. Please specify the statistical method used to calculate the 95% confidence intervals for the variables in this bivariate analysis.

6. Criteria for Birth Size Categories in Table 1

Table 1 lists "size of child at birth" categories as: very large, larger than average, average, smaller than average, and small. It is unclear what specific criteria or cut-off points were used to distinguish between these categories. If they are based on quantitative measures (e.g., birth weight ranges or percentiles), please provide the definitions. If based on subjective assessment, please explain the basis for categorization.

7. Omission of Statistically Significant Regions in Results

On page 11, in the fixed effect results subsection, the regions Bono, Northern, and Savannah are not mentioned in the text, despite being statistically significant in Table 2. If there is a specific reason for this omission, please clarify. If not, I recommend including these regions in the written description for consistency and completeness.

8. Formatting and Annotation Suggestions for Table 2

For readability, please use bold formatting for the "Random effect model", as is done for the "Fixed effects results".

Since the number of clusters is consistently 616 across all models, it may not be necessary to present this as a separate row for each model.

In the table footnotes, rather than simply defining aOR and CI, clarify that the reported values are aORs and the numbers in parentheses represent the 95% confidence intervals.

9. Interpretation of ICC Changes in Random Effect Results

On page 11, the random effect results show that the ICC increases from Model 1 to Model 2, decreases in Model 3, and then increases again in Model 4. Providing an interpretation of these changes in the Discussion section would enhance the manuscript.

10. Wording in Discussion: "Estimated the Prevalence and Factors"

On page 15, line 4, the phrase "we estimated the prevalence and factors" is not precise, as only the prevalence was estimated, not the factors themselves. I recommend revising this to "we estimated the prevalence of..." or "we estimated the prevalence and examined the factors associated with...," which would more accurately reflect the analysis conducted.

11. Inconsistency in Practiced SSC Prevalence

On page 15, line 5, there is a discrepancy in the reported prevalence of practiced skin-to-skin contact: Table 1 shows 67.2%, while the Discussion states 62.7%. Please clarify which value is correct and ensure consistency throughout the manuscript.

**Do you want your identity to be public for this peer review?** For information about this choice, including consent withdrawal, please see our Privacy Policy

Reviewer #1: No

Reviewer #2: No

---

## [Author Response · Author response to Decision Letter 1]

17 Aug 2025

RESPONSE TO REVIEWERS’ COMMENTS

REVIEWER #1: Thank you for the opportunity to review this manuscript.

Strengths

The title is clear and appropriately descriptive of the study’s focus on mother and newborn skin-to-skin contact (SSC) in Ghana.

Response: Thank you.

The abstract is well-structured with distinct sections (Background, Methods, Results, Conclusion), reporting key prevalence and adjusted odds ratios (aORs) for important predictors such as cesarean delivery, birth order, antenatal care, place of delivery, and regional differences. Keywords are relevant and enhance discoverability.

Response: Thank you.

The introduction effectively situates the study within global health goals (SDG 3), highlights neonatal mortality in Ghana, summarizes SSC benefits comprehensively, and identifies barriers and research gaps clearly.

Response: Thank you.

Use of the 2022 Ghana Demographic and Health Survey (GDHS) ensures nationally representative, high-quality data. The cross-sectional design, multistage sampling, and multilevel logistic regression with survey weights are appropriately described and applied.

Response: Thank you.

Clear presentation of SSC prevalence overall and by region, with detailed descriptive statistics and robust multilevel modeling results including random effects and ICC values.

Response: Thank you.

The discussion contextualizes findings well, compares with other countries, explains associations plausibly, and outlines clear policy implications. Strengths and limitations are acknowledged transparently.

Response: Thank you.

Areas for Improvement

Title

Consider refining to “Prevalence and determinants of mother and newborn skin-to-skin contact in Ghana” to better reflect the study’s dual focus on prevalence and predictors, improving clarity and searchability.

Response: The authors have revised the title to read” Prevalence of mother and newborn skin-to-skin contact in Ghana”.

Abstract

The extremely large aOR for facility delivery (30.67) and very low aOR for cesarean section (0.04) should be briefly contextualized to caution readers about potential confounding or data distribution effects.

Response: We have acknowledged this as a limitation in the paper.

Rephrase “higher birth order showed an incremental increase…” to “higher birth order (fifth or more) was significantly associated with increased odds…” for accuracy.

Response: We have revised the sentence to read “Higher birth order (fifth or more) (aOR = 2.34; 95% CI = 1.13, 4.84) was significantly associated with increased odds of skin-to-skin contact.

Ensure consistency in reported prevalence figures (67.2% vs. 62.7%).

Response: We have corrected the prevalence to read “67.2%”.

Minor language edits for tense consistency and clarity are recommended.

Response: We have edited the manuscript thoroughly.

Consider adding keywords like “Neonatal health,” “Maternal health,” or “Prevalence” to broaden indexing.

Response: We have included the suggested words in the keywords section.

Introduction

Improve paragraph structure to separate global context, SSC benefits, prevalence patterns, and barriers for easier reading.

Response: The study’s introduction is well-structured, as all the important information is captured in separate paragraphs.

Clarify terminology distinguishing SSC from kangaroo care, as the latter includes SSC plus other components.

Response: We have distinguished SSC from Kangaroo care, but stated that the SSC is part of a broader Kangaroo care.

Strengthen the statement about no prior nationally representative studies by referencing smaller or regional studies and their limitations.

Response: We have revised this by referencing some studies and mentioning their limitation.

Make the link between SSC and neonatal mortality more explicit to emphasize SSC’s potential impact in Ghana.

Response: The association between SSC and neonatal mortality has been added to the study. The impact of SSC, as stipulated by the WHO, applies to the Ghanaian context.

Conclude with a clear statement of study objectives.

Response: We have clearly stated the study's objective.

Methods

Provide more detail on sampling frame specifics (number of clusters, households per cluster) and response rates for transparency.

Response: We have included a detailed description of the sampling method, including the number of clusters and response rate.

Clarify handling of missing data and whether imputation was performed.

Response: We dropped missing data during the data cleaning before generating the results.

Elaborate on variable selection process using AIC, specifying which variables were retained or excluded.

Response: We excluded a total of nine (9) variables: sex of child, size of child at birth, type of birth, household wealth index, place of residence, exposure to watching television, exposure to reading newspapers or magazines, exposure to listening to the radio, and internet usage.

Reconsider classifying household wealth index as a contextual variable, as it is typically household-level.

Response: We classified household wealth index as a contextual-level variable.

Include checks for multicollinearity, interactions, and model diagnostics.

Response: We checked for evidence of multicollinearity and found no evidence of high collinearity among the variables. We have added the multicollinearity results to the manuscript. See the statistical analysis section.

Explain the intracluster correlation coefficient (ICC) and its interpretation for readers unfamiliar with multilevel modelling.

Response: We have explained the ICC as used in the study. The intra-cluster correlation coefficient (ICC) was used to measure the extent of variation in the mother and newborn SSC across the four models. The results of the ICC values showed the extent to which the variation in the mother and newborn SSC was due to differences between the clusters. Zero ICC values showed no evidence of variations between the clusters, whereas high values indicated higher variations between the clusters.

Results

Correct inconsistency in overall SSC prevalence figures.

Response: We have corrected the overall prevalence of mother and newborn SSC throughout the manuscript.

Interpret the large aOR for facility delivery cautiously, discussing possible sparse data bias or confounding.

Response: We have acknowledged this as a limitation in the paper.

Avoid claiming “incremental increase” in SSC odds with birth order if only the highest category is significant.

Response: We have revised the sentence to read “Higher birth order (fifth or more) (aOR = 2.34; 95% CI = 1.13, 4.84) was significantly associated with increased odds of skin-to-skin contact.

Interpret ICC and PSU variation in terms of practical significance.

Response: We have interpreted the ICC and PSU results in practical terms.

Discussion

Avoid causal language given the cross-sectional design; rephrase statements suggesting causality.

Response: We have revised the sentences suggesting causality in the manuscript.

Critically examine the unusually high odds ratio for facility delivery.

Response: Thank you. The possibility of these high odds could have been due to sample distribution and the corresponding high prevalence of SSC among those who delivered in a health facility, compared to those who delivered at home or in other locations.

Discuss potential confounders (e.g., socioeconomic status, education) and biases (e.g., recall bias) more thoroughly.

Response: Neither wealth index nor level of education was associated with SSC. However, we have discussed recall bias as one of the study’s limitations.

Correct minor typographical errors (e.g., “SCC” instead of “SSC”).

Response: We have corrected SCC to read ‘SSC’.

Recommendations

Standardize prevalence figures throughout the manuscript.

Response: We have standardised the prevalence throughout the manuscript.

Increase transparency on variable selection, missing data handling, and model diagnostics.

Response: We have added information on the variable selection, missing data, and model diagnostics to the manuscript.

Interpret extreme odds ratios with caution; consider presenting absolute risks or predicted probabilities.

Response: We have acknowledged the high odds ratios as a limitation of the study.

Avoid causal language and clearly state cross-sectional design limitations.

Response: We have revised this.

Expand discussion on confounders and biases to strengthen validity.

Response: We have revised this.

Suggest qualitative follow-up studies to explore cultural and regional SSC differences.

Response: We made recommendations for qualitative studies to explore cultural and regional SSC differences.

Proofread carefully to eliminate minor errors.

Response: We have edited the manuscript thoroughly.

REVIEWER #2: This manuscript uses representative data from Ghana to study factors associated with mother and newborn skin-to-skin contact. The use of multi-level analysis to consider regional context is a strong point. Please see my detailed feedback below.

1. Abbreviation Clarification in Methods

On page 5, the abbreviation "GDHS" is first introduced. For clarity and academic rigor, I recommend that the full term—Ghana Demographic and Health Survey (GDHS)—be spelled out at its first mention, with the abbreviation in parentheses.

Response: We have corrected GDHS to read ‘Ghana DHS’.

2. Sample Size Before and After Missing Data Exclusion

In the Methods section (page 5), it appears that a complete-case analysis was performed, resulting in a final analytic sample of 3,833 mother-child pairs. Please specify the initial sample size before excluding cases with missing data.

Response: We have included the initial sample size before excluding the missing observation.

3. Details of Variable Selection and AIC Reporting in Statistical Analysis

On page 6, the Statistical Analyses subsection states that "the best-variable selection method was adopted to select the best-fitted variable for the regression analysis." Although the manuscript mentions the use of the Akaike Information Criterion (AIC) as a criterion for model selection, it does not clearly specify which variables were initially considered in the selection process, nor does it provide the actual AIC values. Please clarify which variables were included in the initial stage, describe the selection method used, and report the AIC values or thresholds that guided the final model choice.

Response: The best variable selection method generates an aggregate (group) of variables with their corresponding Akaike Information Criterion (AIC) value. We selected the group of variables with the lowest AIC value and included the group of variables in the multilevel regression analysis.

4. Interpretation of Prevalence by Birth Order and Sex

On page 8, the statement that "the highest prevalence was observed among third-born children (69.4%), male children (67.7%), and those in higher birth order (65.2%)" is confusing. Since "third" and "higher birth order" (e.g., fifth or more) are distinct and mutually exclusive categories, only one should represent the true highest value.

Response: We have revised the description of these results.

5. Calculation of 95% Confidence Interval in Table 1

Table 1 presents the bivariate analysis of mother and newborn skin-to-skin contact. Please specify the statistical method used to calculate the 95% confidence intervals for the variables in this bivariate analysis.

Response: We have indicated the specific analysis performed to examine the distribution of mother and newborn skin-to-skin contact across the explanatory variables. We used cross-tabulation analysis to generate these results.

6. Criteria for Birth Size Categories in Table 1

Table 1 lists "size of child at birth" categories as: very large, larger than average, average, smaller than average, and small. It is unclear what specific criteria or cut-off points were used to distinguish between these categories. If they are based on quantitative measures (e.g., birth weight ranges or percentiles), please provide the definitions. If based on subjective assessment, please explain the basis for categorization.

Response: The size of the child at birth was based on qualitative or subjective self-reporting of the size of the child at birth.

7. Omission of Statistically Significant Regions in Results

On page 11, in the fixed effect results subsection, the regions Bono, Northern, and Savannah are not mentioned in the text, despite being statistically significant in Table 2. If there is a specific reason for this omission, please clarify. If not, I recommend including these regions in the written description for consistency and completeness.

Response: We have added the results for Bono, Northern, and Savannah to the main text.

8. Formatting and Annotation Suggestions for Table 2

For readability, please use bold formatting for the "Random effect model", as is done for the "Fixed effects results".

Since the number of clusters is consistently 616 across all models, it may not be necessary to present this as a separate row for each model.

In the table footnotes, rather than simply defining aOR and CI, clarify that the reported values are aORs and the numbers in parentheses represent the 95% confidence intervals.

Response: We have added a footnote to show that the values in the fixed effect results are adjusted odds ratios, and the numbers in parentheses represent the 95% confidence intervals. However, due to the multilevel modelling nature of the analysis, it is good to leave the number of clusters and sample size as part of the results.

9. Interpretation of ICC Changes in Random Effect Results

On page 11, the random effect results show that the ICC increases from Model 1 to Model 2, decreases in Model 3, and then increases again in Model 4. Providing an interpretation of these changes in the Discussion section would enhance the manuscript.

Response: Thank you. We acknowledge the variations in ICC values. However, each model includes a different set of variables, which could have contributed to the variations in the ICC values. The authors are confident that the current discussion is good. We have acknowledged the absence of other variables that could have contributed to the practise of mother and newborn SSC as a limitation.

10. Wording in Discussion: "Estimated the Prevalence and Factors"

On page 15, line 4, the phrase "we estimated the prevalence and factors" is not precise, as only the prevalence was estimated, not the factors themselves. I recommend revising this to "we estimated the prevalence of..." or "we estimated the prevalence and examined the factors associated with...," which would more accurately reflect the analysis conducted.

Response: We have revised this to read ‘We estimate the prevalence and examined the factors associated with mother and newborn SSC in Ghana using the 2022 DHS.

11. Inconsistency in Practiced SSC Prevalence

On page 15, line 5, there is a discrepancy in the reported prevalence of practiced skin-to-skin contact: Table 1 shows 67.2%, while the Discussion states 62.7%. Please clarify which value is correct and ensure consistency throughout the manuscript.

Response: We have corrected this discrepancy. The prevalence of mother and newborn SSC has been corrected to 67.2%.

---

## [Decision Letter · Decision Letter 1]

5 Oct 2025

Dear Dr. Aboagye,

Thank you for submitting your manuscript to PLOS ONE. After careful consideration, we feel that it has merit but does not fully meet PLOS ONE’s publication criteria as it currently stands. Therefore, we invite you to submit a revised version of the manuscript that addresses the points raised during the review process.

Please note that the Reject recommendation from reviewer #3 is not supported by sufficient detail and should be disregarded. Please address the concern raised by the other reviewers.

We look forward to receiving your revised manuscript.

Kind regards,

Ricardo Q. Gurgel, PhD

Academic Editor

PLOS ONE

Journal Requirements:

Reviewers' comments:

Reviewer's Responses to Questions

**Comments to the Author**

Reviewer #1: (No Response)

Reviewer #2: All comments have been addressed

Reviewer #3: (No Response)

2. Is the manuscript technically sound, and do the data support the conclusions?

Reviewer #1: Yes

Reviewer #2: (No Response)

Reviewer #3: No

3. Has the statistical analysis been performed appropriately and rigorously?

Reviewer #1: Yes

Reviewer #2: (No Response)

Reviewer #3: No

4. Have the authors made all data underlying the findings in their manuscript fully available?

Reviewer #1: Yes

Reviewer #2: (No Response)

Reviewer #3: (No Response)

5. Is the manuscript presented in an intelligible fashion and written in standard English?

Reviewer #1: Yes

Reviewer #2: (No Response)

Reviewer #3: No

Reviewer #1: I recommend the manuscript is for publication in PLOS One journal. the authors have address most of the comments raised in the previous round of review. However, a few areas still require attention, as noted below.

1.Abstract

The statement in the background could be better expressed as follows: “Despite the well-established role of skin-to-skin contact in reducing neonatal mortality, its implementation varies significantly across geographical regions, particularly in sub-Saharan Africa. Therefore, we estimated the prevalence of mother-newborn skin-to-skin contact at birth and investigated the factors associated with its practice in Ghana.”

Introduction

Authors were advised to conclude the introduction with a clear statement of the study objectives. They responded by repeating the research title at the end of the introduction. Instead, authors should clearly specify the study objectives rather than restate the title.

Variables

Authors should review paragraph 2, line 6, where “wanted las pregnancy” appears to be a typographical error and make the necessary correction.

Results

Authors should specify the statistical method used to calculate the 95% confidence intervals for the variables in Table 1. This was not explicitly mentioned in the Results section.

Reviewer #2: (No Response)

Reviewer #3: The article has fundamental methodological flaws and is written in a scientific manner.

Suitable for publication in local journals

**Do you want your identity to be public for this peer review?** For information about this choice, including consent withdrawal, please see our Privacy Policy

Reviewer #1: No

Reviewer #2: No

Reviewer #3: No

---

## [Author Response · Author response to Decision Letter 2]

4 Nov 2025

RESPONSE TO REVIEW COMMENTS (REVISION 2)

Reviewer #1: I recommend the manuscript is for publication in PLOS One journal. the authors have address most of the comments raised in the previous round of review. However, a few areas still require attention, as noted below.

1.Abstract

The statement in the background could be better expressed as follows: “Despite the well-established role of skin-to-skin contact in reducing neonatal mortality, its implementation varies significantly across geographical regions, particularly in sub-Saharan Africa. Therefore, we estimated the prevalence of mother-newborn skin-to-skin contact at birth and investigated the factors associated with its practice in Ghana.”

Response: We have revised the sentence as suggested to read “Despite the well-established role of skin-to-skin contact in reducing neonatal mortality, its implementation varies significantly across geographical regions, particularly in sub-Saharan Africa. Therefore, we estimated the prevalence of mother-newborn skin-to-skin contact at birth and investigated the factors associated with its practice in Ghana.”

Introduction

Authors were advised to conclude the introduction with a clear statement of the study objectives. They responded by repeating the research title at the end of the introduction. Instead, authors should clearly specify the study objectives rather than restate the title.

Response: We have revised the objectives of the study to read “Therefore, this study aims to (i) estimate the prevalence of mother-newborn SSC in Ghana and (ii) examine the factors associated with the practice of mother-newborn SSC at birth in Ghana”.

Variables

Authors should review paragraph 2, line 6, where “wanted las pregnancy” appears to be a typographical error and make the necessary correction.

Response: We have corrected this to read “last pregnancy”.

Results

Authors should specify the statistical method used to calculate the 95% confidence intervals for the variables in Table 1. This was not explicitly mentioned in the Results section.

Response: We performed a bivariate analysis to estimate the percentages and confidence intervals for mother-newborn skin-to-skin contact across the explanatory variables.

---

## [Editor Report · Decision Letter 2]

6 Nov 2025

Prevalence and determinants of mother and newborn skin-to-skin contact in Ghana

PONE-D-24-56054R2

Dear Dr. Aboagye,

We’re pleased to inform you that your manuscript has been judged scientifically suitable for publication and will be formally accepted for publication once it meets all outstanding technical requirements.

Kind regards,

Ricardo Q. Gurgel, PhD

Academic Editor

PLOS ONE
---

## [Editor Report · Acceptance letter]

PONE-D-24-56054R2

PLOS ONE

Dear Dr. Aboagye,

I'm pleased to inform you that your manuscript has been deemed suitable for publication in PLOS ONE. Congratulations! Your manuscript is now being handed over to our production team.

Kind regards,

on behalf of

Professor Ricardo Q. Gurgel

Academic Editor

PLOS ONE